# Textual Training for the Hassle-Free Removal of Unwanted Visual Data : Case Studies on OOD and Hateful Image Detection

**Saehyung Lee**[1]* **Jisoo Mok**[1] **Sangha Park**[1] **Yongho Shin**[2] **Dahuin Jung**[3]† **Sungroh Yoon**[1,4]†

[1]Department of Electrical and Computer Engineering, Seoul National University
[2]Qualcomm Korea YH, Seoul, South Korea
[3]School of Computer Science and Engineering, Soongsil University
[4]Interdisciplinary Program in Artificial Intelligence, Seoul National University
`{halo8218, magicshop1118, wiarae}@snu.ac.kr,`
`yshin@qti.qualcomm.com, dahuin.jung@ssu.ac.kr, sryoon@snu.ac.kr`

## Abstract

In our study, we explore methods for detecting unwanted content lurking in visual datasets. We provide a theoretical analysis demonstrating that a model capable of successfully partitioning visual data can be obtained using only textual data. Based on the analysis, we propose Hassle-Free Textual Training (HFTT), a streamlined method capable of acquiring detectors for unwanted visual content, using only synthetic textual data in conjunction with pre-trained vision-language models. HFTT features an innovative objective function that significantly reduces the necessity for human involvement in data annotation. Furthermore, HFTT employs a clever textual data synthesis method, effectively emulating the integration of unknown visual data distribution into the training process at no extra cost. The unique characteristics of HFTT extend its utility beyond traditional out-of-distribution detection, making it applicable to tasks that address more abstract concepts. We complement our analyses with experiments in out-of-distribution detection and hateful image detection. Our codes are available at `https://github.com/Saehyung-Lee/HFTT`

## 1 Introduction

We are currently in the midst of what is known as the large-scale AI era. The growth in both the size of deep neural networks and training datasets led to unparalleled achievements in a wide array of tasks [4, 43]. However, this transition to large-scale AI presents new, unforeseen challenges. Particularly, the recent reports on the biased behavior of large AI models raise significant concerns surrounding the continuously expanding size of training datasets without proper quality control and regulation [2, 41]. The massive scale of visual training datasets necessary to train large-scale models presents a challenge in curating data to ensure unbiased and safe datasets. This is primarily due to the impracticality of manually selecting and removing unwanted content from such an extensive collection of images. This issue of data curation has traditionally been addressed by: (i) creating a supervised dataset for a specific objective; (ii) training a model on this dataset; and then (iii) utilizing the model to develop a larger dataset [28]. However, this approach exploits considerable human labor and needs to be re-initiated from the beginning whenever there are changes in the training objective.

---

*This publication was created by Saehyung Lee while an intern at QTI and currently attends Seoul National University.
†Corresponding Authors

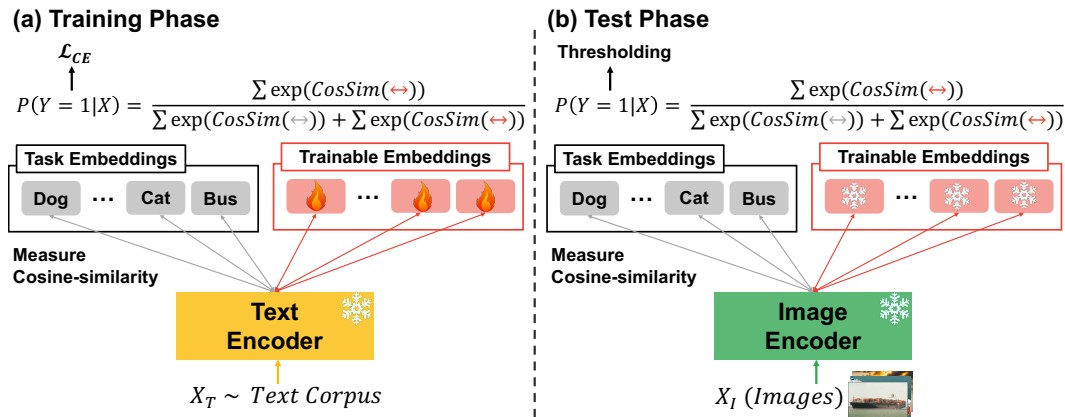

Figure 1: Overview of our proposed method. **Task embeddings** define the task to be performed. For example, in the case of hateful image detection, hate speeches would serve as task embeddings, while in OOD detection, the names of classes from the training distribution would be the task embeddings. **Trainable embeddings** are the only parameters that are trained in our method, defined in the joint embedding space. During the training phase, only textual data are used, and in the testing phase, these trained parameters are employed to classify images. Detailed explanations are provided in Sections 3.

The field of out-of-distribution (OOD) detection, which aims identify OOD data that lie outside the training data distribution, can be considered a sub-branch of data curation research. Recent works in OOD detection utilize vision-language models (VLMs) [40, 29, 30] to take advantage of the rich and human-aligned representations learned by these models. For instance, Esmaeilpour et al. [11] augmented the pre-trained CLIP model [40] with an additional decoder module, trained on a vision-language dataset, for visual OOD detection. In a similar vein, Wang et al. [49] incorporated an extra "no" text encoder, trained on a vision-language dataset, into CLIP. These previous approaches, however, suffer from a significant limitation: They require a vast amount of additional vision-language data. Using data samples targeted for detection can improve sample efficiency, but this leads to the dilemma of needing to collect unwanted data for the purpose of removing them.

In this work, we propose a novel method that no longer relies on additional visual data or a computationally expensive training process. We first outline our theoretical rationale, particularly demonstrating that with a successfully trained model on a bimodal dataset, like CLIP, one can obtain a classifier to partition data from one mode using solely the data from the other mode. Building upon our motivation, we propose a method called Hassle-Free Textual Training (HFTT). HFTT consists of a newly proposed loss and a clever textual data synthesis method, updating trainable parameters defined in the joint embedding space to improve the detection of undesirable visual content. Specifically, we decompose the weighted cross-entropy loss into a formula that includes a regularization term, which is tailored to our use. Additionally, to achieve higher detection accuracy, we introduce the concept of focal loss [33]. Moreover, our textual data synthesis method, which combines prompt templates and words, can effectively imitate the involvement of the entire visual data distribution in the training process. We illustrate an overview of our proposed method in Figure 1.

The proposed loss function brings considerable convenience in achieving our objective. To train an unwanted data detection model, it is necessary to define out-distribution for a given data distribution (in-distribution), which is not always straightforward due to vague boundaries between the two. For instance, in hateful content detection tasks [13], the divide between what is hateful and what is not is influenced by various contexts, *e.g.*, historical and social backgrounds. Our proposed loss eliminates the need for human labor to annotate out-distribution data because it does not involve a clearly defined set of out-distribution data. Furthermore, our textual data synthesis method requires no cost, employing a rule-based approach using only prompt templates and a set of words.

Based on the principle that HFTT can detect out-distribution samples by merely defining the in-distribution in natural language, we propose that this method can be extended to tasks beyond traditional OOD detection, including hateful image detection. Current OOD detection methods often fail in such extended tasks for two main reasons: firstly, they are based on the assumption of a distinct boundary between in- and out-distributions, which is unsuitable for tasks with abstract concepts.

Secondly, methods requiring training images may lead to ethical concerns. Our proposed method, however, is not subject to these limitations.

Through empirical validation, we verify that HFTT can enhance the performance of VLMs in identifying unwanted visual data, whether it be OOD instances or hateful images. Additionally, we demonstrate through feature visualization results that HFTT, despite not observing any visual data in the training phase, appears to have been trained as if it had. Furthermore, we provide various analyses of HFTT, including comparative results of using different textual data synthesis methods.

In summary, our **contributions** are as follows: (i) We theoretically demonstrate how textual data can serve as a substitute for visual data in our scenario; (ii) We introduce a new loss function. Its use eliminates the need for labor in annotating out-distribution data; (iii) We propose a textual data synthesis method that can efficiently imitate the visual data distribution in our training; (iv) We empirically analyze HFTT, a method composed of the above proposals. Our experiments show that HFTT is effective in a range of scenarios, from traditional OOD detection to situations involving abstract concepts, like the identification of hateful images.

## 2 Related Work

**Vision-language models.** With the advancements in deep learning, tackling sophisticated tasks that demand an understanding of both vision and language modalities has become viable. The methodologies employed to encode image and text data exhibit notable distinctions owing to their inherent differences. Prominent within this domain are dual-stream models exemplified by CLIP [40], ALIGN [23], and FILIP [51]. These models employ separate encoders for text and image data, optimizing them through contrastive objectives to align semantically similar features across heterogeneous modalities. Primarily, VLMs integrate transformer-based encoders for text data, while a variety of architectures, encompassing convolutional neural networks [25, 15] and vision transformers [9], are deployed for image encoding. The success of CLIP-like models has spurred numerous subsequent inquiries, with a focus on enhancing data efficiency and adaptability for diverse downstream tasks.

**Out-of-distribution detection.** Traditionally, OOD detection has evolved by defining post-hoc OOD scores [18, 32, 27, 34] or formulating learning algorithms based on outlier exposure methods [26, 20, 10]. With the advancement of VLMs, methods for OOD detection that leverage both image and text embeddings have also progressed. Post-hoc OOD score methods based on VLM typically involve utilizing the OOD class name [12, 11] or defining OOD scores using the top similarity values between images and class names [37]. In the case of outlier exposure, which requires a training algorithm, some approaches employ prompt learning or fine-tune the image encoder [45] of models like CLIP. Transitioning from conventional methods to VLM-based approaches, none of these methods have attempted text-only training and subsequently applied their techniques to tasks such as hateful image detection.

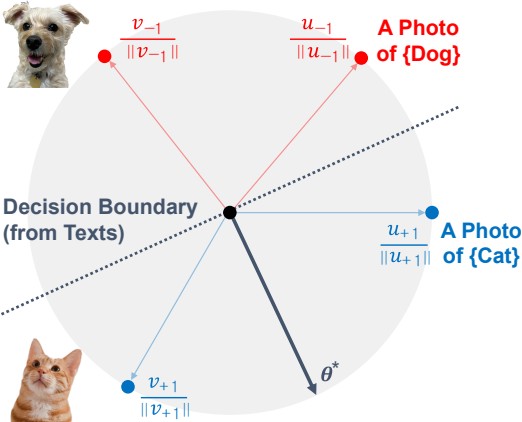

Figure 2: Overview of Section 3.1. The red and blue colors symbolize the two classes $-1$ and $+1$, respectively. In our theoretical model, $u$ and $v$ can be interpreted as text and image, respectively.

**Text-only training for vision tasks.** Given the progress in VLMs, there have been numerous studies aimed at replacing images with textual representations in vision and multimodal tasks. Textual data presents the advantage of being easily collectible compared to visual data. Previous studies have demonstrated the effectiveness of using only textual information for various vision tasks, including image classification [38], image captioning [31], and multi-label image recognition [14]. Our work represents a pioneering effort in applying text-only supervision to unwanted visual data detection.

# 3 Method

In this section, we propose a new textual training method for the convenient and successful removal of unwanted visual data. In Section 3.1, we theoretically demonstrate, through a motivating example, that when there is a well-trained model on a bimodal dataset, such as CLIP, it is possible to train a binary classifier successfully partitioning data from one mode using only data from the other. This theoretical revelation leads us to our novel loss function that allows hassle-free training of an unwanted visual data detector in Section 3.2. Lastly, in Section 3.3, we present our proposed method that includes a simple yet effective synthesis of textual data. The proposed method is executable even without access to the parameters of the backbone model, making it lightweight and applicable to black-box foundational models.

## 3.1 A Motivating Example

We theoretically demonstrate that when there exists a well-trained bimodal model $F : \mathcal{G} \times \mathcal{H} \to \mathcal{Z}$ for a given bimodal data distribution, it is possible to train a classifier successfully partitioning data from one mode ($\mathcal{H}$) using only the dataset from the other mode ($\mathcal{G}$). To align with the operations of VLMs like CLIP in our scenario, we assume that the output vectors of $F$ are normalized. We define a bimodal dataset $D$ as follows:

$$D = \{(g_i, h_i, y_i)\}_{i=1}^{N}, \text{ where } y \overset{u.a.r.}{\sim} \{-1, +1\} \text{ and } (g, h) \overset{i.i.d.}{\sim} \mathcal{G}_y \times \mathcal{H}_y.$$

$(g_i, h_i)$ represents the input vectors from the two modes for a given data sample, where $y_i$ denotes the binary class of the $i$-th sample. We can partition the dataset $D$ as follows:

$$D = D_{-1} \cup D_{+1}, \text{ where } D_y = \{(g_i, h_i) \,|\, y = y_i\}.$$

We assume that samples belonging to the same class in the dataset $D$ exhibit similar semantic patterns. Given $F$ that successfully builds the joint embedding space for the bimodal data distribution, we can posit the following:

$$u_{+1}^{\top} v_{+1} > u_{+1}^{\top} v_{-1} \text{ and } u_{-1}^{\top} v_{+1} < u_{-1}^{\top} v_{-1}, \text{ where } u_y = \mathop{\mathbb{E}}_{u \in U_y} [u] \,, v_y = \mathop{\mathbb{E}}_{v \in V_y} [v] \,,$$

$$U_y = \{F(g_i) \,|\, g_i \in D_y\}, \text{ and } V_y = \{F(h_i) \,|\, h_i \in D_y\}.$$

$U_y$ and $V_y$ are class-conditional embedding sets for each of the two modes, respectively. For simplicity, we assume that the variances of the angular distributions relative to their mean vectors are equal for sets $U_{-1}$ and $U_{+1}$, as well as for sets $V_{-1}$ and $V_{+1}$.

We investigate whether the cosine-similarity classifier $\theta^{\star}$ trained solely on the unimodal dataset $(g_i, y_i)_{i=1}^{N}$ using $F$ can successfully be applied to $(h_i, y_i)_{i=1}^{N}$. We establish the following theorem:

**Theorem 1.** *For the quadratic loss function $L(u, y; \theta) = \left(1 - y\theta^{\top} u\right)^2$, the optimal cosine-similarity classifier $\theta^{\star}$ that classifies sets $U_{-1}$ and $U_{+1}$ is*

$$\arg\min_{\theta} \mathop{\mathbb{E}}_{u \in U_{-1}} [L(u, -1; \theta)] + \mathop{\mathbb{E}}_{u \in U_{+1}} [L(u, +1; \theta)] = \frac{u_{+1} - u_{-1}}{\|u_{+1} - u_{-1}\|}.$$

Proofs are in Appendix A. Theorem 1 demonstrates the optimal classifier $\theta^{\star}$ is orthogonal to $u_{-1} + u_{+1}$. We present an illustration in Figure 2 to enhance understanding of both the problem under investigation and the results of our analysis.

Applying the classifier $\theta^{\star}$ trained to classify $U_{-1}$ and $U_{+1}$ to distinguish between $V_{-1}$ and $V_{+1}$ leads to the following:

**Corollary 1.** *The classifier $\theta^{\star}$, with respect to $V_{-1}$ and $V_{+1}$, satisfies the double inequalities of*

$$\mathop{\mathbb{E}}_{v \in V_{-1}} \left[\theta^{\star \top} v\right] < 0 < \mathop{\mathbb{E}}_{v \in V_{+1}} \left[\theta^{\star \top} v\right].$$

This implies that we can successfully classify $V_{-1}$ and $V_{+1}$ by observing the cosine similarities with $\theta^{\star}$. Motivated by these theoretical examples, we hypothesize that classifiers obtained solely using textual data can operate on visual data as well. Section 4 empirically demonstrates that the arguments developed based on our theoretical model can be applied to modern machine-learning settings.

## 3.2 Our Proposed Loss Function

Our objective is to distinguish in-distribution data samples ($D_{\text{in}}$), conforming to given data distribution, from out-distribution data samples ($D_{\text{out}}$). The development of our new loss function begins with defining the binary cross-entropy loss $L$ as follows:

$$L(u, y) = -\frac{1+y}{2} \log p(u) - \frac{1-y}{2} \log(1 - p(u)).$$

We employ the notations introduced in Section 3.1. $p(u)$ denotes the probability that the label of an embedding $u$ is $+1$, where $+1$ signifies an out-distribution. With respect to datasets $U_{-1}$ and $U_{+1}$, we minimize

$$\sum_{u \in U_{-1}} \lambda L(u, -1) + \sum_{u \in U_{+1}} (1 - \lambda) L(u, +1). \tag{1}$$

We introduce a hyper-parameter, $\lambda \in [0, 1]$, to adjust the balance between in-distribution learning (the first term) and out-distribution learning (the second term). Equation (1) can be reformulated as

$$= \sum_{u \in U_{-1}} L(u, -1) - \sum_{u \in U_{-1}} (1 - \lambda) L(u, -1) + \sum_{u \in U_{+1}} (1 - \lambda) L(u, +1). \tag{2}$$

The second term can be understood as regularization for in-distribution learning. As $\lambda$ approaches 0, in-distribution learning is more heavily impeded. Rather than employing the original regularization term, $-\sum_{u \in U_{-1}} (1 - \lambda) L(u, -1)$, we propose changing it to

$$\sum_{u \in U_{-1}} (1 - \lambda) L(u, +1).$$

Before analyzing the significance of this modification to the objective function, we first examine the effects resulting from this change. Along with the modification, our objective function can be formulated as follows:

$$\sum_{u \in U_{-1}} L(u, -1) + \sum_{u \in U_{-1}} (1 - \lambda) L(u, +1) + \sum_{u \in U_{+1}} (1 - \lambda) L(u, +1)$$
$$= \sum_{u \in U_{-1}} L(u, -1) + \sum_{u \in U_{-1} \cup U_{+1}} (1 - \lambda) L(u, +1). \tag{3}$$

To minimize Eq. 2, it is imperative to distinguish between the in-distribution dataset and the out-distribution dataset. In-distribution data aligns with the objective of the given task, and any data not included in it becomes out-distribution data. However, in real-world scenarios, distinguishing between these distributions is not straightforward. For instance, if we consider $U$ as the text embedding space, collecting out-distribution texts for a given set of in-distribution texts involves considerations such as homonyms, synonyms, and various forms of linguistic variations. Particularly, in tasks where the boundaries between in-distribution and out-distribution are ambiguous, as seen in challenges such as hate content detection [13], constructing a dataset for Eq. 2 becomes difficult and requires considerable human labor. However, the utilization of Eq. 3 alleviates us from such challenges. In other words, the union of the two sets $U_{-1} \cup U_{+1}$ in Eq. 3 allows us to treat all data samples as out-distribution without the need to ponder their relationship with the in-distribution, providing a solution to the intricacies involved in dataset construction. The distinction between Eq. 2 and 3 becomes evident when comparing the gradient signals produced by the two different regularization terms. Gradients of the regularization terms in Eq. 2 and 3 can be computed as follows:

$$\text{(original)} \quad -\sum_{u \in U_{-1}} \frac{\partial L(u, -1)}{\partial p(u)} = \sum_{u \in U_{-1}} \frac{-1}{1 - p(u)},$$

$$\text{(proposed)} \quad \sum_{u \in U_{-1}} \frac{\partial L(u, +1)}{\partial p(u)} = \sum_{u \in U_{-1}} \frac{-1}{p(u)}.$$

The original regularization term weakly regularizes in-distribution samples that were sufficiently learned by the model (*i.e.*, samples with low $p(u)$). However, the proposed regularization term does the exact opposite; it imposes stronger regularization on in-distribution samples with low

$p(u)$. In essence, our proposed regularization prevents the model from exhibiting high confidence in in-distribution samples and enforces the decision boundary to be created near the in-distribution.

Recent studies show that the closer the decision boundary of the out-distribution data detector is to the in-distribution, the more effective the detector is at identifying various out-distribution data [26, 19, 10, 39]. Subsequent research efforts have been directed at obtaining out-distribution samples that reside close to the in-distribution while training a detector to bring its decision boundary closer to the in-distribution. For instance, Lee et al. [26] utilizes a generative adversarial network to acquire samples placed on the in-distribution boundary. Du et al. [10] models the in-distribution using a Gaussian distribution and samples embeddings from the low-likelihood regions of the defined Gaussian distribution. Likewise, we focus on training samples situated in the region close to the in-distribution by incorporating an additional focal loss.

The focal loss was initially proposed to forcefully suppress the gradients for background pixels, which dominate the image, and intensify the learning signals from foreground pixels. Under our scenario, the in-distribution, like foreground pixels, tends to inhabit a small portion of the entire embedding space. In light of the similarity between the in-distribution and foreground pixels, we utilize the focal loss to restrain the loss from far out-distribution samples and amplify learning signals from samples near the in-distribution. The proposed loss can thus be defined as:

**Definition 1.** *Let $B_{-1} = \{x_i\}_{i=1}^N$ and $B = \{\tilde{x}_i\}_{i=1}^N$ denote mini-batches that are respectively drawn from the specified in-distribution and the overall data distribution. Let $L$ be the cross-entropy loss. Then, our proposed loss function is*

$$\sum_{x_i \in B_{-1}} L(x_i, -1) + (1 - \lambda) \sum_{x_j \in B} \beta_j L(x_j, +1); \quad \beta_j = \frac{N\alpha_j}{\sum_{x_k \in B} \alpha_k} \text{ and } \alpha_j = (1 - p(x_j))^\gamma. \quad (4)$$

$p(x)$ is the predictive probability of $x$ belonging in out-distribution. $\gamma \geq 0$ is treated as a hyperparameter of the focal loss. In Section C, we compare the results of using loss terms in Eqs. 2 and 3 and demonstrate the particular effectiveness of the focal loss.

### 3.3   Hassle-Free Textual Training (HFTT)

So far, we have assumed access to data sampled from the out-distribution. However, we may not always be able to anticipate the out-distribution in advance, and even if we can, sampling a subset of data that is representative of the entire distribution is not a straightforward problem in nature. In our scenario, we solely utilize textual data to learn an unwanted visual data detector. Therefore, the proposed scenario requires texts to define the in-distribution and a comprehensive corpus of textual data that can replace the entirety of visual data. VLMs, such as CLIP, obtained impressive zero-shot classification accuracy on diverse visual data benchmarks through the usage of prompts, *e.g.*, "a photo of a { }." Inspired by the success of prompting in VLMs, we conjecture that all visual data can be expressed textually through prompts. This assumption allows the textual dataset to replace the unknown visual data distribution by integrating words associated with the visual data into prompts, drastically simplifying the process of textual data sampling in our method. One example of a prompt design utilized in our approach is: "This is a photo of a { }." To emulate the effect of using the entire visual data distribution, we adopt a word set[3] that includes approximately 370k English words. We report the results of using other prompt designs or textual data acquisition processes in Appendix C. While the optimization procedure in our method additionally requires in-distribution textual data according to Eq. 4, these can be obtained with minimal effort by creating arbitrary sentences or prompts related to the given task. Section 4 details how in-distribution textual data are attained for each experimental setting.   Even though the task of unwanted visual data detection is a type of binary classification problem, learning a plain linear classifier in the embedding space of pre-trained VLMs through approaches like linear probing is not necessarily compatible with our task because the in- and out-distributions are not expected to be linearly separable. To accurately estimate the probability that input $x$ belongs in the out-distribution $p(x)$, we must take advantage of the informative signals in the text encoder of pre-trained VLMs. In our method, $p(x)$ is computed as follows:

1. Obtaining embeddings $\{w_i^{\text{in}}\}_{i=1}^K$ for $K$ texts that effectively represent in-distribution visual data is equivalent to defining the task. This process is akin to obtaining zero-shot classifiers using VLMs. We will refer to these text embeddings as **task embeddings**.

---

[3] https://github.com/dwyl/english-words?tab=readme-ov-file

---

**Algorithm 1** Hassle-Free Textual Training (HFTT)

---

**Require:** word set $\mathcal{W}$, prompt templates $\mathcal{P}$, in-distribution textual data $\mathcal{G}_{-1}$, task embeddings $\{w_i^{\text{in}}\}_{i=1}^{K}$, trainable embeddings $\{w_j^{\text{out}}\}_{j=1}^{N}$, pre-trained model $F$, hyper-parameter $\lambda$

 1: Initialize the trainable embeddings $\{w_j^{\text{out}}\}_{j=1}^{N}$
 2: **for** mini-batches $(B_{-1}, \texttt{words}) \sim (\mathcal{G}_{-1}, \mathcal{W})$ **do**
 3:     $B \leftarrow \texttt{word2data}(\texttt{words}, \mathcal{P})$     # textual data synthesis (the entire data distribution)
 4:     Compute the proposed loss by Eq. 4
 5:     Update $\{w_j^{\text{out}}\}_{j=1}^{N}$     # the incurred cost is negligible
 6: **end for**
 7: **Output:** out-distribution data detector $\left(F, \{w_i^{\text{in}}\}_{i=1}^{K}, \{w_j^{\text{out}}\}_{j=1}^{N}\right)$

---

2. With a pre-trained vision-language model $F$ and the set of $N$ **trainable embeddings** $\{w_j^{\text{out}}\}_{j=1}^{N}$ defined in the joint embedding space, $p(x)$ is obtained as

$$\frac{\sum_{j=1}^{N} \exp\left(F(x)^{\top} w_j^{\text{out}}\right)}{\sum_{i=1}^{K} \exp\left(F(x)^{\top} w_i^{\text{in}}\right) + \sum_{j=1}^{N} \exp\left(F(x)^{\top} w_j^{\text{out}}\right)}.$$

Our method minimizes the custom loss defined in Eq. 4 by learning $\{w_j^{\text{out}}\}_{j=1}^{N}$ only with textual data with the task embeddings $\{w_i^{\text{in}}\}_{i=1}^{K}$ and the model $F$ kept frozen. Because the trainable embeddings are tuned in the output space of the backbone network, the proposed method results in little memory and computational cost. Furthermore, no need to access the parameters of the backbone network makes the proposed method extensible to black-box foundation models. The overall procedure of the proposed method is summarized in Algorithm 1.

## 4 Experimental Results and Discussion

### 4.1 Experimental Setup

We complement our analysis with case studies conducted on OOD and hateful image detection. For the OOD detection task, ImageNet-1k [7] is treated as in-distribution, and the following datasets are used as out-distribution data: iNaturalist [47], SUN [50], Places [52], and Textures [5]. We specifically utilize OOD datasets that are carefully curated to be disjoint from ImageNet, as described in [22]. For hateful image detection, we utilize a dataset containing 892 Antisemitic/Islamophobic images and 420 phrases (Hate) [13]. The Hate dataset is a human-annotated dataset whose usage is limited to individuals with academic purposes to prevent its unethical and unregulated use.

We adopt CLIP, the most extensively studied VLM, specifically using ViT-B/16 as the vision backbone. Unless specified otherwise, we set the batch size=256, learning rate=1.0, epoch=1, $\gamma$=1.0 (the focal loss hyper-parameter), $\lambda$=0, and $N$=10 (the number of trainable embeddings) for all experiments. Note that in the majority of scenarios, there is a substantial predominance of out-distribution textual data compared to in-distribution textual data, and our approach involves mini-batch sampling. Consequently, given the rarity of in-distribution data sampling, the training on in-distribution data remains largely unaffected even though $\lambda = 0$. All values presented in the tables of this paper are the average results over five runs. We conduct a comparative analysis of our approach against existing methods requiring in-distribution images, namely Mahalanobis [27], MSP [18], KNN [44], and NPOS [46]. Additionally, we include methods that do not necessitate in-distribution data, Energy [34], ZOC [11], MaxLogit [21], and MCM [36], in our comparison. The evaluation is performed using OOD scores proposed by the aforementioned works, as well as the scores introduced in this paper (refer to $p(x)$ in Sec 3.3). The calculation of Area Under the Receiver Operating Characteristic (AUROC) and False Positive Rate at 95% True Positive Rate (FPR95) is based on these scores.

**HFTT training and inference costs.**   The backbone model for HFTT remains untrained, while only trainable parameters (trainable embeddings) defined in the model output space are updated. Thus, the cost of this update process is almost equivalent to the forward propagation cost of the synthetic textual data. For the corpus ($\sim$370k samples) and CLIP utilized in the experiments, the update process takes less than 2 minutes with a single V100 GPU. During inference time, the computation of HFTT

Table 1: Comparison of HFTT and competitive baselines with and without in-distribution image requirements on the ImageNet-1K dataset. The best and second-best results are indicated in bold and underlined, respectively. Our method surpasses even strong baselines that utilize in-distribution images. This complements our analysis in Section 3.1, demonstrating that textual data can substitute for visual data in such tasks.

| Method | OOD Dataset | | | | | | | | | |
| | iNaturalist | | SUN | | Places | | Texture | | Average | |
| | FPR | AUROC | FPR | AUROC | FPR | AUROC | FPR | AUROC | FPR | AUROC |
|---|---|---|---|---|---|---|---|---|---|---|
| **In-distribution images required** | | | | | | | | | | |
| Mahalanobis | 99.33 | 55.89 | 99.41 | 59.94 | 98.54 | 65.96 | 98.46 | 64.23 | 98.94 | 61.51 |
| MSP | 40.17 | 89.76 | 63.99 | 79.40 | 63.50 | 80.19 | 67.01 | 79.33 | 58.67 | 82.17 |
| KNN | 29.17 | 94.52 | 35.62 | 92.67 | **39.61** | **91.02** | 64.35 | 85.67 | 42.19 | 90.97 |
| NPOS | **16.58** | **96.19** | 43.77 | 90.44 | 45.27 | 89.44 | 46.12 | **88.80** | 37.94 | 91.22 |
| **In-distribution images not required** | | | | | | | | | | |
| Energy | 34.70 | 90.55 | 32.33 | 90.58 | 40.29 | 89.32 | 51.24 | 72.36 | 39.64 | 85.70 |
| MaxLogit | 35.03 | 89.46 | 32.86 | 90.33 | 41.15 | 89.60 | 68.17 | 75.63 | 44.30 | 86.26 |
| ZOC | 87.30 | 86.09 | 81.51 | 81.20 | 73.06 | 83.39 | 98.90 | 76.46 | 85.19 | 81.79 |
| MCM | 34.33 | 91.36 | 32.27 | 91.86 | 47.48 | 88.68 | 50.90 | 87.52 | 41.25 | 89.86 |
| **HFTT (ours)** | 27.44 | 93.27 | **19.24** | **95.28** | 43.54 | 90.26 | **43.08** | 88.23 | **33.33** | **91.76** |

involves obtaining cosine similarities between trainable embeddings and input embeddings. This cost amounts to 2×(batch size)×(embedding dimension)×(the number of trainable embeddings) FLOPS, which is negligible compared to the inference cost of the entire model.

## 4.2 Out-of-Distribution Detection

In OOD detection experiments, we utilize weights of zero-shot classifiers of pre-trained VLMs as task embeddings for HFTT. In-distribution textual data are obtained via combinations of various prompt templates and class names of in-distribution data, which, in our experimental setting, are equivalent to 1,000 ImageNet classes. We adopt the prompt set released by OpenAI for prompt ensembling[4] as prompt templates. The comparison results are reported in Table 1. Despite the simple and lightweight nature of the proposed method, it outperforms even strong baselines that utilize images, on average.

To understand how the trained embeddings provide the task embeddings with informative signals for identifying OOD data points, we visually analyze the joint embedding space of CLIP in Figure 3. Even though visual OOD data were not involved in the training process, the trained embeddings are positioned on a sub-region of the embedding space inhabited by OOD data and thus can function as additional pointers for where OOD data lie in the joint embedding space. Therefore, this deliberate positioning of trained embeddings precludes the task embeddings from accidentally confusing out-distribution data as in-distribution by refining the decision boundary to intricately separate in-distribution and out-distribution regions.

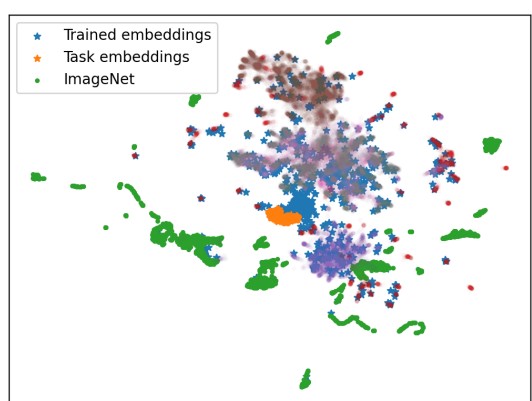

Figure 3: UMAP [35] visualization of the joint embedding space of CLIP. The dispersed, transparent markers represent the OOD data samples used in our experiment (iNaturalist: brown; SUN: grey; Places: pink; Texture: purple; NINCO [3]: red). The trained embeddings (blue stars) are located in a sub-region of the embedding space occupied by OOD data. We trained 2000 embeddings for this plot. It is important to note that these trainable embeddings did not incorporate any information about the OOD data during their training time.

---

[4] https://github.com/openai/CLIP

Table 2: Comparison of HFTT with state-of-the-art methods for OOD detection that do not require in-distribution images, conducted on the Hate dataset. The best result in each column is in bold. HFTT outperforms baseline approaches, showing that it can effectively be used for the general purpose of unwanted data detection.

| Method | iNaturalist | | SUN | | Innocuous Dataset Places | | Texture | | NINCO | | Average | |
|---|---|---|---|---|---|---|---|---|---|---|---|---|
| | FPR | AUROC | FPR | AUROC | FPR | AUROC | FPR | AUROC | FPR | AUROC | FPR | AUROC |
| Energy | 12.30 | 97.53 | 3.88 | 98.51 | 13.87 | 97.40 | 39.70 | 94.98 | 26.89 | 96.12 | 17.43 | 97.10 |
| MaxLogit | 23.65 | 96.89 | 18.49 | 97.49 | 27.84 | 96.48 | 33.33 | 96.00 | 33.03 | 95.99 | 25.82 | 96.71 |
| ZOC | 87.76 | 71.05 | 66.51 | 85.23 | 69.96 | 82.57 | 65.48 | 83.22 | 81.06 | 78.36 | 74.15 | 84.09 |
| MCM | 80.53 | 76.70 | 87.54 | 69.38 | 81.37 | 74.12 | 60.39 | 84.97 | 81.95 | 78.00 | 77.45 | 76.29 |
| CLIPN | 47.71 | 92.78 | 36.36 | 95.16 | 40.62 | 94.52 | 53.36 | 92.36 | 68.40 | 89.58 | 49.29 | 92.88 |
| NegLabel | **0.03** | **99.84** | 1.09 | 99.10 | 5.16 | 98.50 | 3.56 | 98.82 | 12.62 | 97.86 | 4.49 | 98.82 |
| **HFTT (ours)** | 0.17 | 99.44 | **1.05** | **99.13** | **4.38** | **98.60** | **1.73** | **99.08** | **4.18** | **98.52** | **1.83** | **99.06** |

This visualization result can be attributed to two methodological characteristics that are unique to our method. First, our method directly optimizes the trainable embeddings and is not bounded by the modality gap between texts and images. Second, our method successfully places the trained embeddings on top of OOD data only through textual data; this consolidates that textual data can replace visual data, providing strong empirical support for the theory presented in Section 3.1. Together, these two aspects of HFTT yield the joint embedding space as illustrated in Figure 3.

### 4.3 Hateful Image Detection

In this task, the hateful data that contains offensive content against Muslims and Jews is treated as in-distribution data, whereas innocuous data void of such content is treated as out-distribution data. Consequently, embeddings of distinct phrases from a collection of offensive and hateful phrases, provided as part of the Hate dataset, are utilized as task embeddings. The entire set of offensive and hateful phrases is employed as in-distribution textual data.

The Mahalanobis, MSP, KNN, and NPOS methods necessitate the construction of an in-distribution image dataset. Therefore, they should not be used as methods for unethical image detection tasks such as hate image detection, as doing so would require the construction of unethical image datasets, leading to numerous ethical problems such as direct or indirect leakage of sensitive information. In contrast, HFTT requires no usage of any image, thus it can be applied to any unethical image detection task without ethical concerns. To highlight the differences between traditional OOD detection methods and HFTT, we include two additional baselines [49, 24] and one extra dataset [3].

In Table 2, we can observe that most OOD detection methods show significantly lower performance compared to HFTT. These results arise because OOD detection methods assume a classification problem with clear distinctions between classes. In tasks dealing with abstract concepts, the boundaries between data clusters within the in-distribution are ambiguous, which results in the underperformance of existing OOD detection methods. NegLabel shows different results compared to traditional OOD detection methods but still falls short of our proposed approach. We provide a further comparison of our method to CLIPN [49] and NegLabel [24] in Appendix B.

To further study the generalizability of HFTT, we observe the effectiveness of HFTT in low-quality image detection [17] and within the medical image domain [6, 16, 48]. The findings demonstrate the potential extension of HFTT's applicability beyond conventional OOD detection tasks. The results of these experiments and an ablation study on hyper-parameters are provided in Appendices B and C.

## 5 Conclusion and Limitation

In this paper, we proposed a novel methodology for identifying undesirable content hidden within visual datasets. Close theoretical scrutiny of the joint embedding space of VLMs led to the development of HFTT, an efficient framework for training detectors to automatically identify unwanted visual content by leveraging solely textual data together with pre-trained VLMs. HFTT is comprised of a creative objective function that markedly diminishes human involvement in data annotation and the textual data synthesis technique in HFTT that can simulate the usage of unknown visual

data distributions in the training process without additional cost. The distinctive attributes of HFTT broaden its applicability from a clearly-scoped OOD detection task to a far more general set of tasks that are more abstract. Because HFTT requires some type of VLM as the base model, its capabilities are bounded by the representative capacity of pre-trained VLMs. This dependency on pre-trained VLMs makes the use of HFTT in tasks that VLMs struggle with challenging.

**Impact Statements.** This paper contributes to the growing field of data curation and selection research. As datasets for training large AI models expand without adequate safeguards, identifying unwanted data points, such as biased or offensive content, from training datasets is becoming crucial. We believe our work will make a positive contribution to this area, opening up new possibilities for the effortless removal of unwanted visual data. While our method could potentially be misused for content censorship, we believe the positive impact it provides significantly outweighs these concerns.

## Acknowledgments and Disclosure of Funding

This work was supported by Institute of Information & communications Technology Planning & Evaluation (IITP) grant funded by the Korea government (MSIT) [No.RS-2021-II211343, 2022-0-00959, RS-2022-II220959, Artificial Intelligence Graduate School Program (Seoul National University)], the National Research Foundation of Korea (NRF) grant funded by the Korea government (MSIT) (No. 2022R1A3B1077720, 2022R1A5A708390811), and the BK21 FOUR program of the Education and the Research Program for Future ICT Pioneers, Seoul National University in 2024.

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

# A   Proofs

*Theorem 1.* For the quadratic loss function $L\left(u, y; \theta\right) = \left(1 - y\theta^\top u\right)^2$, the optimal cosine-similarity classifier $\theta^\star$ that classifies sets $U_{-1}$ and $U_{+1}$ is

$$\arg\min_\theta \mathbb{E}_{u \in U_{-1}} \left[L\left(u, -1; \theta\right)\right] + \mathbb{E}_{u \in U_{+1}} \left[L\left(u, +1; \theta\right)\right]$$

$$= \arg\min_\theta \left(1 + \theta^\top u_{-1}\right)^2 + \left(1 - \theta^\top u_{+1}\right)^2 = \frac{u_{+1} - u_{-1}}{\|u_{+1} - u_{-1}\|}.$$

*Proof.* Our assumption validates the following equations:

$$\theta^\top \theta = u^\top u = v^\top v = 1, \Sigma_{+1} + \Sigma_{-1} = \epsilon \mathbb{I}, \|u_{+1}\| = \|u_{-1}\|,$$

where $\Sigma_y$ and $\mathbb{I}$ denote the covariance matrix of $U_y$ and the identity matrix, respectively, and $\epsilon > 0$ is a constant. Then,

$$\arg\min_\theta \mathbb{E}_{u \in U_{-1}} \left[L\left(u, -1; \theta\right)\right] + \mathbb{E}_{u \in U_{+1}} \left[L\left(u, +1; \theta\right)\right]$$

$$= \arg\min_\theta \mathbb{E}_{u \in U_{-1}} \left[\left(1 + \theta^\top u\right)^2\right] + \mathbb{E}_{u \in U_{+1}} \left[\left(1 - \theta^\top u\right)^2\right]$$

$$= \arg\min_\theta \left(\mathbb{E}_{u \in U_{-1}} \left[1 + \theta^\top u\right]\right)^2 + \left(\mathbb{E}_{u \in U_{+1}} \left[1 - \theta^\top u\right]\right)^2 + \theta^\top \Sigma_{+1} \theta + \theta^\top \Sigma_{-1} \theta$$

$$= \arg\min_\theta \left(\mathbb{E}_{u \in U_{-1}} \left[1 + \theta^\top u\right]\right)^2 + \left(\mathbb{E}_{u \in U_{+1}} \left[1 - \theta^\top u\right]\right)^2 + 2\epsilon$$

$$= \arg\min_\theta \left(1 + \theta^\top u_{-1}\right)^2 + \left(1 - \theta^\top u_{+1}\right)^2.$$

The gradient of the objective function with respect to $\theta$ is

$$-2\left(1 - \theta^\top u_{-1}\right) u_{-1} + 2\left(1 + \theta^\top u_{+1}\right) u_{+1}.$$

Therefore, the optimal cosine-similarity classifier $\theta^\star$ satisfies the following equation:

$$\left(1 - \theta^{\star\top} u_{-1}\right) u_{-1} = \left(1 + \theta^{\star\top} u_{+1}\right) u_{+1}.$$

Then,

$$1 - \theta^{\star\top} u_{-1} = 1 + \theta^{\star\top} u_{+1} \text{ or } 1 - \theta^{\star\top} u_{-1} = -1 - \theta^{\star\top} u_{+1}.$$

The second equation is not true for any $u_{-1}$ and $u_{+1}$. Hence,

$$\theta^{\star\top} = \frac{u_{+1} - u_{-1}}{\|u_{+1} - u_{-1}\|}.$$

$\square$

*Corollary 1.* The classifier $\theta^\star$, with respect to $V_{-1}$ and $V_{+1}$, satisfies the double inequalities of

$$\mathbb{E}_{v \in V_{-1}} \left[\theta^{\star\top} v\right] < 0 < \mathbb{E}_{v \in V_{+1}} \left[\theta^{\star\top} v\right].$$

*Proof.* Based on the inequalities $u_{+1}^\top v_{+1} > u_{+1}^\top v_{-1}$ and $u_{-1}^\top v_{+1} < u_{-1}^\top v_{-1}$,

$$\mathbb{E}_{v \in V_{-1}} \left[\theta^{\star\top} v\right] = \frac{u_{+1}^\top v_{-1} - u_{-1}^\top v_{-1}}{\|u_{+1} - u_{-1}\|} < 0 < \mathbb{E}_{v \in V_{+1}} \left[\theta^{\star\top} v\right] = \frac{u_{+1}^\top v_{+1} - u_{-1}^\top v_{+1}}{\|u_{+1} - u_{-1}\|}.$$

$\square$

# B   Additional Experiments

**Low-quality image detection.**   We additionally demonstrate the applicability of our method, HFTT, in detecting low-quality images, which are commonly unwanted visual data beyond OOD and hate images. Specifically, we assume the task of detecting corrupted images lurking within a raw visual dataset consisting of 1000 ImageNet classes. For this experiment, we employ ImageNet and ImageNet-C as in-distribution and out-distribution data, respectively. As shown in Table 3, HFTT consistently surpasses existing methods in the detection of corrupted images.

Table 3: Comparison of HFTT with baselines on the low-quality image detection.

| Method | FPR | AUROC |
|---|---|---|
| MSP | 64.17 | 83.94 |
| Energy | 99.99 | 09.16 |
| MaxLogit | 78.01 | 68.47 |
| MCM | 51.54 | 89.06 |
| HFTT (ours) | **42.13** | **92.81** |

Table 4: Comparison of HFTT with MCM on the medical image datasets.

| Method | OOD Dataset | | | |
|---|---|---|---|---|
| | PVQA | | PCAM | |
| | FPR | AUROC | FPR | AUROC |
| MCM | 95.44 | 47.10 | 71.49 | 68.74 |
| MCM + description | 86.84 | 60.39 | 84.50 | 43.44 |
| HFTT | 22.58 | 93.60 | 8.07 | 96.94 |
| HFTT + description | 13.72 | 97.05 | 4.95 | 98.35 |
| HFTT + description + corpus engineering | 6.24 | 98.69 | 4.33 | 98.73 |

**Medical image domain.** We compare the performance of HFTT and MCM in the medical image domain. Specifically, we treat the ISIC-18 skin lesion diagnosis dataset [6] as in-distribution and the PathVQA [16] and PatchCamelyon [48] datasets as out-distribution. The ISIC-18 skin lesion diagnosis dataset is an image classification benchmark for seven skin disease categories. We apply MCM and HFTT to CLIP on the seven disease categories. Table 4 reveals a significantly low detection performance of MCM, attributed to the limited medical domain knowledge of CLIP. Even appending descriptions (generated by GPT-4) to the disease names does not yield favorable results for MCM (+ description). In contrast, our proposed method achieves significantly better results by leveraging the model knowledge and the medical-related information in the corpus. The utilization of medical-related information within the corpus by HFTT is evidenced by the fact that further improvements can be achieved by modifying the corpus to align with the medical domain (+ corpus engineering).

**The experimental results on other pre-trained models.** As discussed in Section 3.1 of our paper, our method assumes that text and image are well-aligned through contrastive learning, similar to CLIP. Therefore, if CLIP is used as the vision encoder, the text encoder must also be CLIP. If text and image are well-aligned, our method can be applied to models other than CLIP. Here, we provide the results for CLIP-L/14, BLIP-B/16, and BLIP-L/16 in addition to the CLIP-B/16 used in our study. Table 5 demonstrates that our method is effective across various vision-language models.

**HFTT vs. CLIPN and NegLabel.** NegLabel [24] constructs an OOD corpus by selecting texts distant from the in-distribution texts from a predefined corpus, then compares the distances between the input image and those texts in the CLIP embedding space to detect OOD. While NegLabel shows high OOD detection performance on ImageNet (see Table 7), it has the following limitations compared to our method: 1) Although NegLabel does not require training additional parameters, it must compute the embeddings of all texts in the corpus and measure their similarity to the in-distribution texts to find the optimal OOD corpus for a given in-distribution. Our training method also requires nearly the same cost as obtaining the embeddings of all texts within a predefined corpus and calculating the similarities between those embeddings and the task+trainable embeddings, as discussed in Section 4.1. Thus, NegLabel and our method require the same level of optimization cost; 2) Since NegLabel uses the embeddings of the determined OOD corpus as they are, it falls behind our method, which has trainable parameters, in terms of generalization. To demonstrate this, we further compare our method and NegLabel in the medical image domain. Specifically, we treat the ISIC-18 skin lesion diagnosis dataset [1] as in-distribution and the PathVQA [16] and PatchCamelyon [48] datasets as out-of-distribution. The ISIC-18 skin lesion diagnosis dataset is an image classification benchmark for seven skin disease categories. Table 6 illustrates the limitations of NegLabel in terms of generalization. While NegLabel fails to construct an effective OOD corpus for the medical image

Table 5: Comparison of HFTT and competitive baselines on the ImageNet-1K dataset. The best result in each column is in bold. Our method outperforms all baselines on both variants of CLIP and BLIP, demonstrating that it can be used to improve the OOD detection performance of various VLMs.

| Model | Method | iNaturalist | | SUN | | OOD Dataset Places | | Texture | | NINCO | |
|---|---|---|---|---|---|---|---|---|---|---|---|
| | | FPR | AUROC | FPR | AUROC | FPR | AUROC | FPR | AUROC | FPR | AUROC |
| CLIP-B | MSP | 34.63 | 91.35 | 32.06 | 91.86 | 47.62 | 88.86 | 49.78 | 87.65 | 72.83 | 71.98 |
| | Energy | 34.70 | 90.55 | 32.33 | 90.58 | 40.29 | 89.32 | 51.24 | 72.36 | 70.06 | 73.85 |
| | MaxLogit | 35.03 | 89.46 | 32.86 | 90.33 | 41.15 | 89.60 | 68.17 | 75.63 | 68.96 | 74.24 |
| | MCM | 34.33 | 91.36 | 32.27 | 91.86 | 47.48 | 88.68 | 50.90 | 87.52 | 73.26 | 71.98 |
| | **HFTT (ours)** | **27.32** | **93.28** | **19.68** | **95.20** | **43.24** | **90.32** | **43.26** | **88.20** | **70.08** | **74.61** |
| CLIP-L | MSP | 26.66 | 94.20 | 22.37 | 94.37 | 36.82 | 92.45 | 52.83 | 86.57 | 67.27 | 78.70 |
| | Energy | 30.84 | 91.25 | 25.94 | 94.10 | 32.94 | 92.30 | 64.33 | 79.26 | 63.49 | 79.72 |
| | MaxLogit | 32.76 | 90.96 | 26.48 | 92.96 | **31.88** | 92.39 | 72.08 | 73.85 | **60.67** | **81.07** |
| | MCM | 26.96 | 94.19 | 22.77 | 94.37 | 36.74 | 92.44 | 52.66 | 86.56 | 68.16 | 78.65 |
| | **HFTT (ours)** | **24.10** | **94.58** | **17.80** | **95.39** | 33.83 | **93.09** | 52.06 | 86.58 | 69.19 | 78.98 |
| BLIP-B | MSP | 64.70 | 82.22 | 30.38 | 91.06 | 71.40 | 78.82 | 76.99 | 81.30 | **71.47** | 72.07 |
| | Energy | 67.15 | 79.30 | 45.21 | 89.07 | 70.28 | 77.49 | 91.24 | 75.38 | 80.29 | **77.20** |
| | MaxLogit | 69.57 | 75.44 | 69.57 | 71.19 | 69.86 | 76.26 | 93.55 | 60.31 | 88.58 | 56.37 |
| | MCM | 64.41 | **82.29** | 30.21 | 91.05 | 70.53 | 79.32 | 75.84 | 81.55 | 71.56 | 72.02 |
| | **HFTT (ours)** | **63.28** | 82.22 | **19.16** | **95.12** | **68.48** | **79.50** | **63.74** | **84.53** | 72.12 | 73.86 |
| BLIP-L | MSP | 51.20 | 87.91 | 22.37 | 93.86 | 61.63 | 84.68 | 64.85 | 85.28 | 65.96 | 78.29 |
| | Energy | 45.63 | 87.23 | 33.94 | 90.29 | 55.73 | 85.91 | 72.38 | 82.16 | 71.23 | 77.49 |
| | MaxLogit | 44.59 | 86.94 | 35.56 | 86.45 | **50.96** | **86.46** | 86.38 | 71.22 | 79.78 | 67.59 |
| | MCM | 50.75 | 88.03 | 22.34 | 93.88 | 60.88 | 85.38 | 64.71 | **85.39** | 66.04 | 78.32 |
| | **HFTT (ours)** | **44.24** | **89.88** | **6.81** | **98.40** | 62.20 | 84.16 | **63.35** | 83.39 | **64.82** | **80.46** |

Table 6: OOD detection in the medical image domain.

| Method | PVQA | | PCAM | |
|---|---|---|---|---|
| | FPR | AUROC | FPR | AUROC |
| CLIPN | 35.47 | 84.64 | 3.10 | 98.76 |
| NegLabel | 37.44 | 94.11 | 48.07 | 94.86 |
| HFTT (ours) | 13.72 | 97.05 | 4.95 | 98.35 |

dataset, our method achieves significantly higher performance by learning optimal embeddings for detection.

CLIPN utilizes an additional "no" text encoder alongside CLIP. This additional text encoder predicts the probability that a given object is not present in an image. Thus, CLIPN predicts whether a given image is in-distribution or out-distribution by using the original CLIP text encoder and the "no" text encoder to estimate the probabilities, respectively. Images with a low probability of being in-distribution and a high probability of being out-distribution are identified as OOD.

Although CLIPN achieves high OOD detection performance on ImageNet (see Table 7), it has the following limitations compared to our method:

1) CLIPN requires significantly higher inference costs due to the use of an additional text encoder; 2) While our method requires lightweight training that does not involve images, CLIPN demands extensive and expensive training of the "no" text encoder on large vision-language datasets; 3) CLIPN can only be applied to tasks where the distinction between in-distribution and out-distribution is clear and straightforward, such as classification datasets. This is because all training images must be classified as either "yes" or "no" images. Therefore, it is unsuitable for tasks dealing with abstract concepts, such as hateful image detection, as discussed in Section 4.3 of our paper; 4) Our method can be easily applied to any detection task defined in natural language, whereas CLIPN shows significantly degraded performance for in-distribution tasks that fall outside the training distribution of the "no" text encoder. In terms of applicability, our proposed method surpasses CLIPN. To demonstrate this,

Table 7: OOD detection performance on ImageNet in-distribution (average for Texture, Places, SUN, and iNaturalist).

| Method | FPR | AUROC |
|---|---|---|
| CLIPN | 31.10 | 93.10 |
| NegLabel | 25.40 | 94.21 |
| HFTT (ours) | 33.33 | 91.76 |

Table 8: Results of using different textual data synthesis methods. HFTT outperforms other methods that are more complex.

| Method | iNaturalist | | SUN | | OOD Dataset Places | | Texture | | NINCO | | Average | |
|---|---|---|---|---|---|---|---|---|---|---|---|---|
| | FPR | AUROC | FPR | AUROC | FPR | AUROC | FPR | AUROC | FPR | AUROC | FPR | AUROC |
| WTO | 29.26 | 92.23 | 23.31 | 93.93 | 40.18 | 90.77 | 42.96 | 88.01 | 69.07 | 76.54 | 40.96 | 88.30 |
| CTO | 27.53 | 91.66 | 38.41 | 86.78 | 43.31 | 89.31 | 60.13 | 79.64 | 76.25 | 69.08 | 49.13 | 83.29 |
| DTO | 29.32 | 92.32 | 24.57 | 94.04 | 43.37 | 89.63 | 42.26 | 89.34 | 70.36 | 74.76 | 41.98 | 88.02 |
| Caption | 54.07 | 79.49 | 57.15 | 74.23 | 63.44 | 77.04 | 41.21 | 91.12 | 89.30 | 56.42 | 47.65 | 86.28 |
| Dedupl. | 28.17 | 93.03 | 21.08 | 94.75 | 43.67 | 90.10 | 42.69 | 88.54 | 68.86 | 75.22 | 40.89 | 88.33 |
| Ours | 27.44 | 93.27 | 19.24 | 95.28 | 43.54 | 90.26 | 43.08 | 88.23 | 70.15 | 74.48 | 40.69 | 88.30 |

we further compare our method with CLIPN in the medical image domain. Table A illustrates the limitations of CLIPN in terms of generalization. While CLIPN effectively detects PCAM, it exhibits very low detection performance on PVQA. In contrast, our method achieves high performance on both OOD tasks.

## C   Ablation Study

In this section, we analyze how different components affect the performance of HFTT.

**Textual data synthesis method.**   While we propose a textual data synthesis method that incurs no additional cost, alternative approaches beyond this can also be explored. We apply the following methods in conjunction with HFTT and list their results in Table 8:

1. WTO, CTO, DTO: Recently, Park *et al.* proposed a method that utilizes textual outliers instead of visual outliers in outlier exposure [19]. For our experiments, we use word-level textual outliers (WTO) generated using in-distribution images along with CLIP and BERT [8], caption-level textual outliers (CTO) generated by an image captioning model [30], and description-level textual outliers (DTO) created using a large language model.
2. Caption: We can consider the extensive use of image captions from LAION-400M [42] to substitute for the entire visual data distribution.
3. Deduplication: To experimentally compare Eq. 2 and 3, we applied HFTT after removing words from our word set that have meanings identical to ImageNet classes as much as possible.

Table 8 reveals that textual outliers generated using additional models and in-distribution images show comparable or inferior performance to our textual data synthesis method. Furthermore, the captions results suggest that heavily relying on image captions does not effectively enhance the average detection performance for various OOD data. Lastly, there appears to be no discernible performance difference between the application of Eq. 1 and 4. This indicates that our proposed loss minimizes the need for human labor by eliminating the process of selecting out-of-distribution data without sacrificing performance.

**The focal loss hyper-parameter.**   HFTT incorporates the concept of focal loss to shape the decision boundary of detectors near the in-distribution. We observe its effect by incrementally increasing the focal loss hyper-parameter $\gamma$ from zero. Table 9 demonstrates that using the focal loss ($\gamma > 0$) generally leads to better performance compared to not using it ($\gamma = 0$).

Table 9: Results of using different values of $\gamma$ for the focal loss. The performance of HFTT appears to be relatively robust to changes in the choice of $\gamma$, with the adoption of focal loss with $\gamma > 0$ generally leading to improved results.

| $\gamma$ | iNaturalist | | SUN | | OOD Dataset Places | | Texture | | NINCO | | Average | |
|---|---|---|---|---|---|---|---|---|---|---|---|---|
| | FPR | AUROC | FPR | AUROC | FPR | AUROC | FPR | AUROC | FPR | AUROC | FPR | AUROC |
| 0 | 27.67 | 93.08 | 20.62 | 94.98 | 43.97 | 90.12 | 44.42 | 87.57 | 69.23 | 75.22 | 41.18 | 88.19 |
| 1 | 27.44 | 93.27 | 19.24 | 95.28 | 43.54 | 90.26 | 43.08 | 88.23 | 70.15 | 74.48 | 40.69 | 88.30 |
| 2 | 27.10 | 93.32 | 19.48 | 95.20 | 43.17 | 90.32 | 42.95 | 88.33 | 70.19 | 74.40 | 40.58 | 88.31 |
| 3 | 27.03 | 93.32 | 19.56 | 95.17 | 43.21 | 90.32 | 42.82 | 88.38 | 70.42 | 74.26 | 40.61 | 88.29 |

Table 10: Results of changing the temperature of the final Softmax layer.

| Temp. | iNaturalist | | SUN | | OOD Dataset Places | | Texture | | NINCO | | Average | |
|---|---|---|---|---|---|---|---|---|---|---|---|---|
| | FPR | AUROC | FPR | AUROC | FPR | AUROC | FPR | AUROC | FPR | AUROC | FPR | AUROC |
| 1.0 | 94.97 | 46.92 | 76.17 | 48.52 | 93.00 | 55.90 | 99.80 | 23.96 | 94.96 | 56.42 | 91.78 | 46.34 |
| 0.1 | 93.52 | 50.17 | 95.22 | 51.30 | 91.67 | 58.43 | 99.75 | 25.37 | 94.62 | 56.93 | 94.96 | 48.44 |
| 0.01 | 27.44 | 93.27 | 19.24 | 95.28 | 43.54 | 90.26 | 43.08 | 88.23 | 70.15 | 74.48 | 40.69 | 88.30 |

**Temperature.** In HFTT, a temperature parameter is utilized for computing $p(x)$. CLIP learns a temperature parameter during their pre-training phase, and we employ these same learned temperature values in all of our experiments. Table 10 illustrates that modifying the temperature value may reduce the efficacy of HFTT.

**The number of trainable embeddings.** If the dimensionality of the data manifold in the joint embedding space of VLMs is low, HFTT can be effective even with a small number of trainable embeddings. To validate this, we present the OOD detection performance of HFTT in Table 11, illustrating how it varies with the number of trainable embeddings ($N$). Remarkably, HFTT can improve OOD detection performance on ImageNet, which has 1,000 classes, even with a very limited number of trainable embeddings. This is possible due to the inherently low dimension of data in the actual model output space [1].

Table 11: Results of shrinking or expanding the number of trainable embeddings ($N$).

| $N$ | iNaturalist | | SUN | | OOD Dataset Places | | Texture | | NINCO | | Average | |
|---|---|---|---|---|---|---|---|---|---|---|---|---|
| | FPR | AUROC | FPR | AUROC | FPR | AUROC | FPR | AUROC | FPR | AUROC | FPR | AUROC |
| 5 | 27.50 | 93.26 | 19.96 | 95.07 | 43.55 | 90.25 | 43.07 | 88.23 | 70.15 | 74.38 | 40.85 | 88.24 |
| 10 | 27.44 | 93.27 | 19.24 | 95.28 | 43.54 | 90.26 | 43.08 | 88.23 | 70.15 | 74.48 | 40.69 | 88.30 |
| 100 | 27.40 | 93.30 | 19.63 | 95.14 | 43.67 | 90.28 | 42.93 | 88.27 | 70.14 | 74.53 | 40.75 | 88.30 |
| 2000 | 27.32 | 93.28 | 19.68 | 95.20 | 43.24 | 90.32 | 43.26 | 88.20 | 70.08 | 74.61 | 40.72 | 88.32 |

