# OpenReview forum: "Textual Training for the Hassle-Free Removal of Unwanted Visual Data: Case Studies on OOD and Hateful Image Detection"
_NeurIPS.cc/2024/Conference — NeurIPS 2024 poster_

### Official Review · Reviewer_kqnV · 2024-07-04

**Soundness:** 3
**Presentation:** 3
**Contribution:** 3
**Rating:** 6
**Confidence:** 3

**Summary:**

This paper proposes a purely textual-based training method for detecting out-of-distribution or hateful images. The authors train an additional embedding layer over frozen CLIP encoders using text data. The authors propose the use of a novel loss function for this training. The method improves performance over baselines for most of the datasets that they evaluate on.

**Strengths:**

Using only textual data for hate detection training is an interesting approach as it eliminates the need to source hateful images or ethical issues of creating paired image-caption datasets of hateful scenarios. The authors provide sufficient experimentation and ablation analysis to validate their claims that only training on text data is enough to identify OOD or hate content. Implementation details are listed and code is provided for reproducibility.

**Weaknesses:**

The writing structure could be improved. The textual synthesis aspect is not clear to me and should be discussed in more detail in the main paper. The method could also be used for generic image classification tasks which brings me to question the choice of why such classification results have not been shown. Some more discussion over the baselines would be good too.

**Questions:**

Clarifications to my questions in weaknesses section is sufficient

**Limitations:**

The authors have described limitations.

---

> ### Author Rebuttal · Authors · 2024-08-07
>
> Firstly, we thank you for your thorough review of our paper. We particularly appreciate your recognition of our innovative approach using only textual data for hate detection, which eliminates the need to source hateful images and addresses ethical concerns. Additionally, we value your acknowledgment of our comprehensive experimentation, ablation analysis, and the provision of implementation details and code for reproducibility.
>
> We are also immensely grateful for your excellent suggestions that can enhance the contents of our paper. We will address each of your concerns in detail in the following:
>
> >**Q1**. The writing structure could be improved. The textual synthesis aspect is not clear to me and should be discussed in more detail in the main paper.
>
> **A1**. We apologize for any inconvenience caused in understanding our paper. In our method, the purpose of textual data synthesis is to emulate the entire visual dataset for training. We assumed that using prompt templates, such as "a photo of a {}", to generate text data could effectively emulate visual data, based on observations of high CLIP zero-shot classification performance with similar prompts. Therefore, our textual data synthesis involves simply inserting numerous words into the prompt "This is a photo of a {}". In our experiments, we used approximately 370k predefined English words. Our results demonstrate that even this lightweight textual data synthesis method can produce a detector capable of identifying various types of unwanted images.
>
> >**Q2**. The method could also be used for generic image classification tasks which brings me to question the choice of why such classification results have not been shown.
>
> **A2**. In fact, our proposed method can always perform generic image classification and unwanted data detection simultaneously. Figure 1 in our paper or the figure in our repository (https://github.com/HFTT-anonymous/HFTT) can aid in understanding this. HFTT utilizes the input embeddings and logit values as they are for zero-shot classification while simultaneously estimating the probability that an input sample is unwanted. For example, our method can be applied to solve a 1001-class classification task consisting of 1000 ImageNet classes plus one OOD class.
>
> However, applying our method does not enhance the zero-shot image classification performance of VLMs. This is because our method focuses solely on optimizing trainable embeddings for out-distribution detection while keeping CLIP frozen. Consequently, the image classification performance of VLMs remains the same as their zero-shot classification performance even after applying our method. Hence, we did not present classification results. For instance, the ImageNet classification accuracy of CLIP-ViT-B/16 remains at 68.6%, regardless of whether our method is applied.
>
> >**Q3**. Some more discussion over the baselines would be good too.
>
> **A3**. Thank you for your suggestion. We provide further discussion on the strongest baselines and present results for various foundation models here:
>
> **Comparison with SOTA OOD detection methods**
> NegLabel [1] and CLIPN [2] are cutting-edge OOD detection methods.
> NegLabel constructs an OOD corpus by selecting texts distant from the in-distribution texts from a predefined corpus, then compares the distances between the input image and those texts in the CLIP embedding space to detect OOD. While NegLabel shows high OOD detection performance on ImageNet (see Table B), it has the following limitations compared to our method:
> - Although NegLabel does not require training additional parameters, it must compute the embeddings of all texts in the corpus and measure their similarity to the in-distribution texts to find the optimal OOD corpus for a given in-distribution. Our training method also requires nearly the same cost as obtaining the embeddings of all texts within a predefined corpus and calculating the similarities between those embeddings and the task+trainable embeddings, as discussed in Section 4.1. Thus, NegLabel and our method require the same level of optimization cost.
> - Since NegLabel uses the embeddings of the determined OOD corpus as they are, it falls behind our method, which has trainable parameters, in terms of generalization. To demonstrate this, we further compare our method and NegLabel in the medical image domain. Specifically, we treat the ISIC-18 skin lesion diagnosis dataset [3] as in-distribution and the PathVQA [4] and PatchCamelyon [5] datasets as out-of-distribution. The ISIC-18 skin lesion diagnosis dataset is an image classification benchmark for seven skin disease categories.
>
> Table A: OOD detection in the medical image domain
> |OOD:|PVQA||PCAM||
> |-|-|-|-|-|
> |**Method**|**FPR** &#8595;|**AUROC** &#8593;|**FPR** &#8595;|**AUROC** &#8593;|
> |NegLabel|37.44|94.11|48.07|94.86|
> |CLIPN|35.47|84.64|3.10|98.76|
> |HFTT (ours)|13.72|97.05|4.95|98.35|
>
> Table A illustrates the limitations of NegLabel in terms of generalization. While NegLabel fails to construct an effective OOD corpus for the medical image dataset, our method achieves significantly higher performance by learning optimal embeddings for detection.
>
> CLIPN utilizes an additional "no" text encoder alongside CLIP. This additional text encoder predicts the probability that a given object is not present in an image. Thus, CLIPN predicts whether a given image is in-distribution or out-distribution by using the original CLIP text encoder and the "no" text encoder to estimate the probabilities, respectively. Images with a low probability of being in-distribution and a high probability of being out-distribution are identified as OOD.
>
> Although CLIPN achieves high OOD detection performance on ImageNet (see Table B), it has the following limitations compared to our method:

---

> > ### Comment · Reviewer_kqnV · 2024-08-09
> >
> > I thank the authors for their rebuttal. The rebuttal answers my concerns, thus I am raising my score. I would suggest the authors to include these discussions in the paper.

---

### Official Review · Reviewer_QxHY · 2024-07-09

**Soundness:** 3
**Presentation:** 3
**Contribution:** 3
**Rating:** 6
**Confidence:** 4

**Summary:**

This paper introduces an efficient and effective text-only training method for detecting undesired visual content. Its key contributions include a theoretical demonstration of how text data can substitute visual data, a new loss function, and a method for synthesizing textual data. These efforts aim to segregate data from one mode using only the dataset from the other mode. The method is evaluated through experiments on OOD detection tasks and hateful image detection, demonstrating comparable performance.

**Strengths:**

1. The utilization of text-only mode for detecting data in the other mode is a compelling approach, supported by existing research in various domains.
2. The training and inference processes are efficient, with a small number of trainable parameters and minimal additional computational costs during inference.
3. Despite the experimental results not consistently outperforming other methods, the high efficiency (no use of image datasets during training) can offset this limitation.

**Weaknesses:**

1. The trainable embedding learning process is unclear. According to the description, trainable embeddings are learned for N out-distribution data instances, which are then frozen during the test phase. It appears that the image embeddings obtained by the image encoder align with the frozen embeddings. If this is the case, the alignment process should be clearly elucidated. If not, the method for using the learned embeddings for images needs to be explained. The parameter N, representing the number of out-distribution data used during training, intuitively suggests that a larger N would enhance accurate distribution learning. However, the results in Table 8 present a contradictory viewpoint. More explanation should be given.
2. The proposed HFTT method can be applied to different pretrained models and OOD tasks. In addition to the image CLIP model, it is recommended that the authors conduct experiments using pretrained video-text models to assess performance changes in related video detection tasks. This additional experiment can shed light on how pretrained models influence performance when only one mode is utilized for learning, considering the crucial role of pre-learned cross-modal alignment knowledge in enabling text-only learning.

**Questions:**

My main questions are (1) making the trainable embedding learning clearer and (2) the test on other pre-trained models to view how the  pre-learned cross-modal alignment knowledge impacts the text-only learning.

**Limitations:**

The main limitations of this paper also lie in the two concerns: unclear description of the trainable embedding learning and no examination of other pre-trained vision/video-text models.

---

> ### Author Rebuttal · Authors · 2024-08-07
>
> Firstly, we thank you for your thorough review of our paper. We particularly appreciate your recognition that our method is the compelling approach of utilizing text-only mode for detecting data in other modes and that the training and inference processes are highly efficient. We are also immensely grateful for your excellent suggestions that can enhance the contents of our paper. We will address each of your concerns in detail in the following:
>
> >**Q1**. The trainable embedding learning process is unclear. According to the description, trainable embeddings are learned for N out-distribution data instances, which are then frozen during the test phase. It appears that the image embeddings obtained by the image encoder align with the frozen embeddings. If this is the case, the alignment process should be clearly elucidated. If not, the method for using the learned embeddings for images needs to be explained.
>
> **A1**. In Section 3.1 of our paper, we presented a motivating example demonstrating that a classifier obtained using only textual data in the output space of a model like CLIP, where images and texts are well-aligned, can also be applied to image data. Based on this, we proceed as follows:
> 1. We define trainable embeddings in the output space of CLIP that serve as the parameters of a classifier distinguishing between in-distribution and out-distribution data.
> 2. We train these embeddings using only textual data.
> 3. We use these trained embeddings to detect unwanted images.
> In summary, the model parameters of CLIP remain fixed, and our trainable parameters are defined and trained within CLIP’s (image-text) joint embedding space. Thus, our method does not require any additional alignment process.
>
> >**Q2**. The parameter N, representing the number of out-distribution data used during training, intuitively suggests that a larger N would enhance accurate distribution learning. However, the results in Table 8 present a contradictory viewpoint. More explanation should be given.
>
> **A2**. We apologize for any confusion caused by our notation. The "N" in Table 8 refers to the number of trainable embeddings, not the number of out-distribution data. The number of trainable embeddings in our method represents the complexity of the classifier that distinguishes between in-distribution and out-distribution data. If the data in CLIP's output space form a very low-dimensional manifold, it would be possible to distinguish between in- and out-distribution data with a small number of trainable embeddings. As we can see in Table 8, a large "N" is not necessary for high unwanted image detection results. Indeed, various studies support the notion that deep learning models possess low-dimensional data manifolds [1,2].
> [1] Ansuini et al. "Intrinsic dimension of data representations in deep neural networks." NeurIPS 2019.
> [2] Moayeri et al. "Text-to-concept (and back) via cross-model alignment." ICML 2023.
>
> >**Q3**. The test on other pre-trained models to view how the pre-learned cross-modal alignment knowledge impacts the text-only learning.
>
> **A3**. Thank you very much for suggesting an expansion of our research domain. Video embedding models and detection tasks are less established than those in the image domain. We will investigate the experimental environment further and include results in the next version of our paper. Here, we provide the results for CLIP-L/14, BLIP-B/16, and BLIP-L/16 in addition to the CLIP-B/16 used in our study. Tables A, B, and C demonstrate that our method is effective across various vision-language models.
>
> Table A: The experimental results on CLIP-L/14 (in-distribution: ImageNet)
> |OOD:|iNaturalist||SUN||Places||Texture||NINCO||
> |-|-|-|-|-|-|-|-|-|-|-|
> |**Method**|**FPR** &#8595; |**AUROC** &#8593;|**FPR** &#8595;|**AUROC** &#8593;|**FPR** &#8595;| **AUROC** &#8593; |**FPR** &#8595;|**AUROC** &#8593;|**FPR** &#8595;|**AUROC** &#8593;|
> |MSP|26.66|94.20|22.37|94.37|36.82|92.45|52.83|86.57|67.27|78.70|
> |Energy|30.84|91.25|25.94|94.10|32.94|92.30|64.33|79.26|63.49|79.72|
> |MaxLogit|32.76|90.96|26.48|92.96|**31.88**|92.39|72.08|73.85|**60.67**|**81.07**|
> |MCM|64.41|26.96|94.19|22.77|94.37|36.74|92.44|52.66|86.56|68.16|78.65|
> |HFTT (ours)|**24.10**|**94.58**|**17.80**|**95.39**|33.83|**93.09**|**52.06**|**86.58**|69.19|78.98|
>
> Table B: The experimental results on BLIP-B/16 (in-distribution: ImageNet)
> |OOD:|iNaturalist||SUN||Places||Texture||NINCO||
> |-|-|-|-|-|-|-|-|-|-|-|
> |**Method**|**FPR** &#8595; | **AUROC** &#8593; |**FPR** &#8595; | **AUROC** &#8593; |**FPR** &#8595; | **AUROC** &#8593; |**FPR** &#8595; | **AUROC** &#8593; |**FPR** &#8595; | **AUROC** &#8593; |
> | MSP |  64.70 | 82.22 | 30.38 | 91.06 | 71.40 | 78.82 | 76.99 | 81.30 | **71.47** | 72.07   |
> | Energy | 67.15 |  79.30 |  45.21 |  89.07 |  70.28  | 77.49 |  91.24 |  75.38 |  80.29  | **77.20**  |
> | MaxLogit |  69.57 | 75.44 | 69.57 | 71.19 | 69.86 | 76.26 | 93.55 | 60.31 | 88.58 | 56.37 |
> | MCM| 64.41  | **82.29** | 30.21 | 91.05 | 70.53 | 79.32 | 75.84 | 81.55 | 71.56 | 72.02  |
> | HFTT (ours)    | **63.28** | 82.22 | **19.16** | **95.12** | **68.48** | **79.50** | **63.74** | **84.53** | 72.12 | 73.86   |
>
> Table C: The experimental results on BLIP-L/16 (in-distribution: ImageNet)
> | OOD:  | iNaturalist || SUN ||  Places || Texture || NINCO ||
> |-|-|-|-|-|-|-|-|-|-|-|
> |**Method**|**FPR** &#8595;| **AUROC** &#8593; |**FPR** &#8595; | **AUROC** &#8593; |**FPR** &#8595; | **AUROC** &#8593; |**FPR** &#8595; | **AUROC** &#8593; |**FPR** &#8595; | **AUROC** &#8593; |
> | MSP |   51.20| 87.91| 22.37| 93.86| 61.63| 84.68| 64.85| 85.28| 65.96| 78.29 |
> | Energy | 45.63 |87.23 |33.94 |90.29 |55.73 |85.91 |72.38 |82.16 |71.23| 77.49  |
> | MaxLogit |  44.59| 86.94| 35.56| 86.45| **50.96**| **86.46**| 86.38| 71.22| 79.78| 67.59  |
> | MCM| 50.75 |88.03 |22.34 |93.88 |60.88 |85.38 |64.71 |**85.39** |66.04| 78.32  |
> | HFTT (ours)  | **44.24**| **89.88**| **6.81**| **98.40**| 62.20| 84.16| **63.35**| 83.39| **64.82**| **80.46** |

---

> > ### Comment · Reviewer_QxHY · 2024-08-12
> > **replying to the response**
> >
> > Thanks for the response. It well addressed my concern-1. For my concern-2, they promise to include the results in their next version. So, I keep my rating Weak Accept.

---

### Official Review · Reviewer_LurW · 2024-07-12

**Soundness:** 3
**Presentation:** 4
**Contribution:** 3
**Rating:** 6
**Confidence:** 3

**Summary:**

This paper focuses on textual training methods to remove undesirable (such as biased or offensive) visual content and proposes a method for detecting unwanted visual content using only synthetic textual data to partition visual data. The classifier trained on textual content can be successfully transferred to visual content. The method consists of an objective function and a textual data synthesis method. The design of the loss does not require data annotation and the textual data synthesis method can emulate unknown visual data distribution into the training process with no extra cost. The proposed method was proven to be effective for out-of-distribution detection and hateful image detection.

**Strengths:**

1. This paper is clear and well-written.

2. The idea is innovative and solid with theoretical results. The proposed method is simple but effective and can be applied to even black-box foundation models.

3. The experiments are comprehensive and the results showed the effectiveness of this method.

4. The training and testing costs are discussed in this paper which could benefit the application of the proposed method.

**Weaknesses:**

I do not see a significant weakness in the paper since it’s well-organized. Several clarification questions are listed below.

**Questions:**

1. This idea of training on textual content for transferring to visual content is interesting, could the authors provide more insights on more examples of applications of this method except for the OOD detection and hateful image detection? Is it possible to apply it to more binary classification tasks?
2. What encoders are used to obtain the task embedding? The authors mentioned using CLIP as the vision backbone, and is the text encoder during training also from CLIP?
3. How do you decide the number of task embeddings?
4. The paper mentions using the proposed method to detect unwanted content and uses two tasks (OOD detection and hateful image detection. If I understand correctly, my key concern is that the focus of this paper is closer to a novel method for binary classification with an application that classifies unwanted content. Could the authors share more thoughts regarding this? If so, I would suggest slightly paraphrasing the introduction to better reflect the contribution and indicate more potential applications.
5. Can this method be extended to multi-class classifications?

**Limitations:**

The authors adequately addressed the limitations.

---

> ### Author Rebuttal · Authors · 2024-08-06
>
> Firstly, we thank you for your thorough review of our paper. We particularly appreciate your recognition that our paper has no significant weaknesses and is well-organized. We have made our best efforts to address your remaining concerns as follows. If our responses meet your expectations, we would be grateful if you could consider reflecting this in your rating:
>
> >**Q1**. This idea of training on textual content for transferring to visual content is interesting, could the authors provide more insights on more examples of applications of this method except for the OOD detection and hateful image detection? Is it possible to apply it to more binary classification tasks?
>
> **A1**. Our method can be applied to more binary classification tasks. We additionally demonstrate the applicability of our method, HFTT, in detecting low-quality images, which are commonly unwanted visual data beyond OOD and hate images. Specifically, we assume the task of detecting corrupted images lurking within a raw visual dataset consisting of 1000 ImageNet classes. For this experiment, we use ImageNet and ImageNet-C. As shown in Table A below, HFTT consistently surpasses existing methods in the detection of corrupted images.
>
> Table A: Corrupted image detection
> |Method|FPR &#8595;|AUROC &#8593;|
> |-|-|-|
> |MSP|64.17|83.94|
> |Energy|99.99|09.16|
> |MaxLogit|78.01|68.47|
> |MCM|51.54|89.06|
> |HFTT (ours)|**42.13**|**92.81**|
>
> >**Q2**. What encoders are used to obtain the task embedding? The authors mentioned using CLIP as the vision backbone, and is the text encoder during training also from CLIP?
>
> **A2**. As discussed in Section 3.1 of our paper, our method assumes that text and image are well-aligned through contrastive learning, similar to CLIP. Therefore, if CLIP is used as the vision encoder, the text encoder must also be CLIP. If text and image are well-aligned, our method can be applied to models other than CLIP. To demonstrate this, we additionally apply our method to BLIP and observe the results. Table B shows that our method is also effective for BLIP.
>
> Table B: The experimental results on BLIP (in-distribution: ImageNet)
> |OOD:|iNaturalist||SUN||Places||Texture||NINCO||
> |-|-|-|-|-|-|-|-|-|-|-|
> |**Method**|**FPR** &#8595;|**AUROC** &#8593;|**FPR** &#8595;|**AUROC** &#8593;|**FPR** &#8595;|**AUROC** &#8593; |**FPR** &#8595;|**AUROC** &#8593;|**FPR** &#8595;|**AUROC** &#8593;|
> |MSP|64.70|82.22|30.38|91.06|71.40|78.82|76.99|81.30|**71.47**|72.07|
> |Energy | 67.15 |  79.30 |  45.21 |  89.07 |  70.28  | 77.49 |  91.24 |  75.38 |  80.29  | **77.20**  |
> |MaxLogit |69.57 | 75.44 | 69.57 | 71.19 | 69.86 | 76.26 | 93.55 | 60.31 | 88.58 | 56.37 |
> |MCM| 64.41| **82.29** | 30.21 | 91.05 | 70.53 | 79.32 | 75.84 | 81.55 | 71.56 | 72.02  |
> |HFTT (ours)| **63.28** | 82.22 | **19.16** | **95.12** | **68.48** | **79.50** | **63.74** | **84.53** | 72.12 | 73.86   |
>
> >**Q3**. How do you decide the number of task embeddings?
>
> **A3**. The number of task embeddings is not a hyper-parameter but is determined by the in-distribution task. For example, if the in-distribution is ImageNet, the task embeddings are the text embeddings for the 1,000 ImageNet classes, such as ["a photo of a tench", ..., "a photo of a toilet paper"]. In the case of hateful image detection, we used representative hate phrases defined by the dataset as task embeddings. The number of trainable embeddings in our method is a hyper-parameter. We present an ablation study on this in Table 8 of Appendix C, and the results show that our method is not sensitive to this parameter.
>
> >**Q4**. The paper mentions using the proposed method to detect unwanted content and uses two tasks (OOD detection and hateful image detection. If I understand correctly, my key concern is that the focus of this paper is closer to a novel method for binary classification with an application that classifies unwanted content. Could the authors share more thoughts regarding this? If so, I would suggest slightly paraphrasing the introduction to better reflect the contribution and indicate more potential applications.
>
> **A4**. We are hesitant to consider our method as a novel approach for binary classification. Generally, binary classification is defined as the task of finding patterns that best distinguish between two different classes. However, in our scenario, we assume that only one (in-distribution) of the two classes has patterns, while we make no assumptions about the other class to cover all possible cases. Therefore, for binary classification tasks where both classes have specific patterns, our method may not lead to a good solution compared to methods that consider patterns of both classes. Thus, we would like to distinguish our method from general binary classification methods. We appreciate the reviewer for prompting us to view our method in this light.
>
> >**Q5**. Can this method be extended to multi-class classifications?
>
> **A5**. In fact, our proposed method can always perform multi-class classification and unwanted data detection simultaneously. Figure 1 in our paper or the figure in our repository (https://github.com/HFTT-anonymous/HFTT) can aid in understanding this. HFTT utilizes the input embeddings and logit values as they are for zero-shot classification while simultaneously estimating the probability that an input sample is unwanted. For example, our method can be applied to solve a 1001-class classification task consisting of 1000 ImageNet classes plus one OOD class.
>
> However, applying our method does not enhance the zero-shot image classification performance of VLMs. This is because our method focuses solely on optimizing trainable embeddings for out-distribution detection while keeping CLIP frozen. Consequently, the image classification performance of VLMs remains the same as their zero-shot classification performance even after applying our method. For instance, the ImageNet classification accuracy of CLIP-ViT-B/16 remains at 68.6%, regardless of whether our method is applied.

---

> > ### Comment · Reviewer_LurW · 2024-08-11
> >
> > I appreciate the detailed response from the authors and all my concerns are addressed. I will increase my rating from 5 to 6 and suggest that the authors include the information in the revised paper.

---

### Official Review · Reviewer_CzuR · 2024-07-16

**Soundness:** 3
**Presentation:** 3
**Contribution:** 3
**Rating:** 6
**Confidence:** 3

**Summary:**

This paper proposes an objective function for CLIP-based architecture to enhance out-of-distribution (OOD) detection. Instead of relying on OOD image data, the approach extracts OOD words from various sources and updates some trainable embeddings using predefined text embedding. Results show that the proposed approach outperforms previous methods that do not require in-distribution images and has comparable performance with methods that require in-distribution images.

**Strengths:**

\+ This paper shows that is possible to learn OOD without the need of in-distribution images, which I think it is quite valuable and useful in practice

\+ The justification of the proposed idea seems well motivated and clearly presented

**Weaknesses:**

\- The related work section lacks comprehensiveness. While the authors mention some CLIP-based methods, they overlook approaches closely related to their own. Notably, NegLabel (ICLR2024) and CLIPN (ICCV2023) also use textual features for OOD detection and are very similar to the proposed method. The authors should include these works in their related work section, provide benchmarks, and critically analyze the advantages of their approach compared to these existing methods.

\- In the experimental results, authors compare results with CLIPN and NegLabel only in Tab. 2, while they are not mentioned in Tab.1. Also, those two methods are not even cited nor explained in related work. I believe it might have been a last minute effort.

\- Some methods in the literature do not require any training (e.g. NegLabel). Given that the proposed approach involves additional training (even though without images), it is crucial to highlight the differences with these training-free methods.

**Questions:**

Authors should explain why the comparison with NegLabel and CLIPN is partial and the method are not even cited by the paper.

In addition authors should highlight advantages and disadvantages of their method which is in-domain image free versus approaches that are training free, such as NegLabel.

**Limitations:**

Authors addressed the limitations.

---

> ### Author Rebuttal · Authors · 2024-08-06
>
> Firstly, we express our gratitude for your thorough review of our manuscript. Particularly, we appreciate your recognition of our method as valuable and useful in practice, as well as your acknowledgment that the justification of our method is well-motivated and clearly presented. Furthermore, we appreciate your invaluable suggestions, which significantly enrich the substance of our work. In the subsequent sections, we meticulously address each of your concerns as follows:
>
> >**W1**. The related work section lacks comprehensiveness. While the authors mention some CLIP-based methods, they overlook approaches closely related to their own. Notably, NegLabel (ICLR2024) and CLIPN (ICCV2023) also use textual features for OOD detection and are very similar to the proposed method. The authors should include these works in their related work section, provide benchmarks, and critically analyze the advantages of their approach compared to these existing methods.
>
> **A1**. Thank you for your feedback on the two related studies. We can compare our method with each of these studies as follows. This discussion will be added to the final version of our paper:
>
> **Comparison with NegLabel**
> NegLabel constructs an OOD corpus by selecting texts distant from the in-distribution texts from a predefined corpus, then compares the distances between the input image and those texts in the CLIP embedding space to detect OOD. While NegLabel shows high OOD detection performance on ImageNet (see Table B), it has the following limitations compared to our method:
> - Although NegLabel does not require training additional parameters, it must compute the embeddings of all texts in the corpus and measure their similarities to the in-distribution texts to find the optimal OOD corpus for a given in-distribution.
> Our training method also requires nearly the same cost as obtaining the embeddings of all texts within a predefined corpus and calculating the similarities between those embeddings and the task+trainable embeddings, as discussed in Section 4.1. Thus, NegLabel and our method require the same level of optimization cost.
> - Since NegLabel uses the embeddings of the determined OOD corpus as they are, it falls behind our method, which has trainable parameters, in terms of generalization. To demonstrate this, we further compare our method and NegLabel in the medical image domain. Specifically, we treat the ISIC-18 skin lesion diagnosis dataset [1] as in-distribution and the PathVQA [2] and PatchCamelyon [3] datasets as out-of-distribution. The ISIC-18 skin lesion diagnosis dataset is an image classification benchmark for seven skin disease categories.
>
> Table A: OOD detection in the medical image domain
> |    OOD:   | PVQA | | PCAM | |
> | ----------------- | ----- | ----- | ----- | ----- |
> | **Method**     | **FPR** &#8595; | **AUROC** &#8593; | **FPR** &#8595; | **AUROC** &#8593; |
> | NegLabel    | 37.44   | 94.11   |48.07   | 94.86   |
> | CLIPN  | 35.47   |  84.64  |3.10   | 98.76   |
> | HFTT (ours)   | 13.72   | 97.05   |4.95   | 98.35   |
>
> Table A illustrates the limitations of NegLabel in terms of generalization. While NegLabel fails to construct an effective OOD corpus for the medical image dataset, our method achieves significantly higher performance by learning optimal embeddings for detection.
>
> **Comparison with CLIPN**
> CLIPN utilizes an additional "no" text encoder alongside CLIP. This additional text encoder predicts the probability that a given object is not present in an image. Thus, CLIPN predicts whether a given image is in-distribution or out-distribution by using the original CLIP text encoder and the "no" text encoder to estimate the probabilities, respectively. Images with a low probability of being in-distribution and a high probability of being out-distribution are identified as OOD.
>
> Although CLIPN achieves high OOD detection performance on ImageNet (see Table B), it has the following limitations compared to our method:
>
> - CLIPN requires significantly higher inference costs due to the use of an additional text encoder.
> - While our method requires lightweight training that does not involve images, CLIPN demands extensive and expensive training of the "no" text encoder on large vision-language datasets.
> - CLIPN can only be applied to tasks where the distinction between in-distribution and out-distribution is clear and straightforward, such as classification datasets. This is because all training images must be classified as either "yes" or "no" images. Therefore, it is unsuitable for tasks dealing with abstract concepts, such as hateful image detection, as discussed in Section 4.3 of our paper.
> - Our method can be easily applied to any detection task defined in natural language, whereas CLIPN shows significantly degraded performance for in-distribution tasks that fall outside the training distribution of the "no" text encoder. In terms of applicability, our proposed method surpasses CLIPN. To demonstrate this, we further compare our method with CLIPN in the medical image domain. Table A illustrates the limitations of CLIPN in terms of generalization. While CLIPN effectively detects PCAM, it exhibits very low detection performance on PVQA. In contrast, our method achieves high performance on both OOD tasks.
>
> Table B: OOD detection performance on ImageNet in-distribution (average for Texture, Places, SUN, and iNaturalist)
> | OOD:  | Average ||
> | ----------------- | ----- | ----- |
> | **Method**            | **FPR** &#8595; | **AUROC** &#8593; |
> | CLIPN    | 31.10   | 93.10 |
> | NegLabel  | 25.40  |  94.21 |
> | HFTT (ours)   | 33.33   | 91.76 |
>
> [1] Codella, Noel, et al. "Skin lesion analysis toward melanoma detection 2018: A challenge hosted by the international skin imaging collaboration (isic)." arXiv 2019.
> [2] He, Xuehai, et al. "Pathvqa: 30000+ questions for medical visual question answering." arXiv 2020.
> [3] Veeling, Bastiaan S., et al. "Rotation equivariant CNNs for digital pathology." MICCAI 2018.

---

> > ### Comment · Reviewer_CzuR · 2024-08-12
> > **I keep my score**
> >
> > Although authors did not answer to my question about the reasons for having only a partial comparison with NegLabel and CLIPN in the main paper, the authors provided a complete and clear comparison with the mentioned methods. Thus I keep my positive score.

---

### Author Rebuttal · Authors · 2024-08-07

We genuinely appreciate the reviewers' dedicated time and their valuable feedback. Before addressing each reviewer's specific concerns in detail below, we would like to summarize here the contributions of our research that have been recognized and the aspects that have been enhanced in our study.
## Reviewer Acknowledgements of Our Paper's Strengths
- The theoretical justification of our proposed method is solid (CzuR, LurW).
- We demonstrate that it is possible to obtain an effective unwanted image detector without using images (CzuR, QxHY, kqnV).
- Our method is innovative, efficient, effective, and useful in practice (CzuR, LurW, QxHY, kqnV).
- The experiments are comprehensive, and the results show the effectiveness of our proposed method (LurW, kqnV).
- For the reproducibility of our method, we provide a cost analysis, implementation details, and code (LurW, kqnV).

## Enhancements in Our Paper
- We provide additional discussion on the comparison with state-of-the-art OOD detection methods.
- We present comparative results against NegLabel and CLIPN in the medical image domain.
- We offer experimental results and discussion regarding other foundation models, such as BLIP.
- We provide a discussion on the application of our method to multi-class classification tasks.

We provide detailed responses to each reviewer's comments below. We have diligently addressed each of the four reviewers' concerns to the best of our abilities, and believe that the results further underscore the contributions of our study. We plan to integrate the reviewers' suggestions into the revisions of the manuscript, as we believe these adjustments will significantly bolster the paper's overall strength.

---

### Decision · Program_Chairs · 2024-09-25

**Decision:**

Accept (poster)

**Comment:**

The paper originally received borderline scores leaning towards acceptance. The authors provided a detailed rebuttal and the paper has been discussed by the reviewers. The reviewers raised some concerns about missing experiments/comparisons (e.g. with NegLabel and CLIPN) that have been mostly addressed in the rebuttal. The authors included in their responses several new result and, overall, the reviewers where satisfied by the rebuttal and they all manteined or increased their scores to 6:WeakAccept. Overall this is a solid submission and the authors are encouraged to carefully follow and implement all the reviewers' suggestion in the final revision of the paper.